# Underwater target depth estimation: A shallow-water broadband acoustic source depth estimation method based on corrected warping transformation

Du Siqi [1], Han Dong [2]*, Liu Cong[3], Li Sidi[1]

1 Midshipmen Group Five, Dalian Naval Academy, Dalian, China, 2 Department of Information System, Dalian Naval Academy, Dalian, China, 3 Unit92896, Dalian Naval Academy, Dalian, China

* small_hand@163.com

**Data Availability Statement:** All relevant data are within the manuscript and its Supporting information files.

## Abstract

A broadband sound source depth estimation method based on the BDRM model is proposed for the shallow sea to address the problem that the traditional Warping transform is limited by the ideal waveguide and cannot handle the real marine environment localization of structures with variable sound speed profiles. The modified Warping transform operator compensates for the propagation and reflection phases of the received signals in double phase, compresses the modes of the received signals under different sound speed profiles to approximate a single frequency, and projects the modified signals using time-frequency analysis (TFR). As a result, the modes of the received signals are clearly separated in the time-frequency domain. Then, the corresponding bandpass filters are designed considering the different marine environments. After that, the modal energies are extracted, and the depth estimation function is constructed based on the matched modal energies to achieve the depth estimation of the pulsed signal target under the non-ideal waveguide conditions with a variable speed of sound profile. Compared to the short-time Fourier transform (STFT) and the dispersion-dissipation transform (DDT), the modified Warping transform (MWT) achieves a clearer separation of the modes of the target received signal under the variable sound speed profile of the non-ideal waveguide. The depth estimation method proposed in this study effectively resolves the issues of passive source depth estimation under variable sound speed profiles, which typically result in a significant decrease in estimation accuracy. Simulation results indicate that under the conditions of the Pekeris waveguide, negative sound speed gradient waveguide, and Qingdao shallow sea waveguide, the success rate of the method for target depth estimation reaches more than 95% under a signal-to-noise ratio (SNR) of 10 dB. This demonstrates both high accuracy and stability. In Qingdao, the target depth can be estimated accurately when the signal-to-noise ratio is above 5 dB. In real sea area experiments, the method effectively separates the first four orders of the normal modes and achieves the estimation of the depth of the airgun pulse source. Compared to the traditional Warping transform, the method proposed in this study provides a wider application range and greater practical value in engineering.

**Funding:** The author(s) received no specific funding for this work.

**Competing interests:** The authors declare that they have no conflict of interest.

## Introduction

In the field of underwater sound source localization research, the key target parameter of depth is of vital importance. Underwater target depth estimation technology holds significant value in raising marine scientific research, marine engineering development, fishery management, and marine resource exploration. With the continuous development and exploration of the marine field, underwater target depth estimation technology is also advancing to meet increasing demands.

The most classic underwater target depth estimation method is the matched field positioning (MFP) technique, which was first proposed by Bucker [1] in 1976. This method matches the actual received acoustic field data with the acoustic field data calculated by the propagation model to determine the most probable position of the acoustic source, realizing the estimation of target depth. Since then, many scholars have improved and developed MFP in terms of environmental matching degree and algorithmic volume [2–6]. However, overcoming the significant degradation of matching accuracy caused by environmental mismatches remains difficult. The modal matching technique (MMP) proposed by Shang [7] is used to address this issue. MMP utilizes the characteristics of each mode in the acoustic field to match the actual received modal structure, achieving target depth estimation. Many scholars have conducted in-depth research and improvements on the matched-mode technique, particularly from the perspectives of filtering and matching algorithms [8–10]. Modal information extraction typically employs a vertical array that spans the entire depth of the water body. Although MMP is robust to environmental mismatches to a certain extent, similar to MFP, it also requires a vertical array for signal acquisition and imposes high requirements on the position of array elements. The vertical array is easily affected by ocean currents and is difficult to maintain, significantly impacting depth estimation. In contrast, the horizontal array is easier to maintain and more feasible for engineering applications [11–14]. However, it is limited by the frequency distribution of the detected target. The implementation of target depth estimation requires a larger number of arrays and longer arrays, which undoubtedly increases the limitations and challenges in practical application scenarios to meet aperture requirements.

Considering the application scenarios and the costs associated with the deployment of sensors at sea, the challenge of underwater acoustics is to develop various underwater technologies with a minimum number of sensors. Compared to time-frequency analysis (TFR)—based underwater depth estimation of targets from a single hydrophone [15], this approach has a better application prospect. In contrast to correlated MMP and uncorrelated MMP, Le Touzé et al. [16, 17] proposed a single hydrophone underwater acoustic source depth estimation method, which obtained more satisfactory results in simulation experiments and validation with real datasets. Due to different geological configurations, time-frequency uncertainties, and interferences between structures, the conventional TFR cannot effectively separate the modes of all orders of the simple normal wave. For this reason, the dissipative dispersive transform (DDT) and the Warping transform, two good time-frequency analysis methods, have been proposed and widely studied [18–20]. Frequency dispersion occurs whenever propagation in a waveguide is involved. The multiple paths of the hydroacoustic channel and the dispersion phenomenon of normal mode theory describe the same physical phenomenon. Solving the dispersion problem also addresses the multiple paths interference problem. The advantage of DDT is that it can cancel the dispersion of the modes of each order within one transformation. The process of dissipating the dispersion is equivalent to modal inverse propagation, which enhances the signals of each order of the modes while suppressing noise and improving the signal-to-noise ratio. Since 2009, Ning Wang and Dazhi Gao from Ocean University of China [21–24] have proposed shallow-sea waveguide Dissipative Dispersion

Processing (DDT) that can simultaneously cancel multimode dispersion based on the expression of waveguide invariants. Their work has been reported at international academic conferences numerous times. Like the DDT, the Warping transform is also a good method for time-frequency analysis, which is divided into the frequency-domain Warping transform and time-domain Warping transform. The frequency-domain Warping transform is similar to the DDT but can only process a single frequency at a time. In contrast, the time-domain Warping transform essentially processes the phase of the received signal. The modes of each order of the received signal can be compressed to a single-frequency signal that approximates its cutoff frequency by resampling the Warping transform. This enables the clear separation of the modes of each order in the time-frequency domain and facilitates the extraction and recovery of single-order modes. The traditional Warping transform is defined in the Pekeris waveguide. Based on the properties of the Warping transform, Guo and Ma [25] separated and extracted the modes from the exploding bomb data of a sea experiment and obtained more satisfactory results. Although the Warping transform is robust, its transform operator is based on Pekeris waveguide projection. However, most actual sea areas are non-ideal waveguides. Under such conditions, the frequency aggregation of the signal processed by the traditional Warping transform is reduced, and the corresponding modal separation ability is also affected. Accordingly, Chi et al. [26] used the autocorrelation function of the received signal to eliminate the influence of marine environmental factors in the Warping transform and effectively processed pulsed acoustic signals in the North Yellow Sea. However, because this method involves autocorrelation of the processed signals rather than direct processing of the received signals, it sacrifices the ability to separate modes of all orders of the normal mode, making subsequent modal extraction impossible. The time-frequency analysis method combining waveguide invariance and Warping transform has been proposed by Li-Cheng and Li [27], Guo et al. [28], and others. The Warping transform combined with waveguide invariants can be applied to non-Pekeris waveguides, making it universally applicable. However, estimating waveguide invariants in different marine environments and assessing the accuracy of the estimation remain significant challenges. Zhang and Li and Zhang and Lu [29, 30] introduced the BDRM (Broadband Depth-Range Modulated) model theory, which can be easily extended to elastic seabeds. In 2014, they proposed a modified Warping transform operator for shallow-sea Pekeris waveguides [31, 32], which successfully counteracted the dispersion of the modes of all orders in simulation experiments. Li [33] and Li et al. [34], based on the BDRM theory, modified the Warping transform to make it more applicable to the liquid version of the semi-infinite seafloor waveguide. Based on this modification, they proposed a ranging method for the target sound source.

The study proposes a single-hydrophone underwater target depth estimation method based on the Warping transformation, grounded in normal mode theory. Based on the BDRM model, the Warping transform operator is corrected and improved by combining the known environmental parameters. The corrected operator is incorporated into the normal mode acoustic field to derive a new acoustic field expression based on the corrected Warping transform acoustic field formula of the BDRM model. The received original target signal is projected into the time-frequency domain through the corrected Warping transform, enabling the modes of the received signal to be realized in the time-frequency domain. The original target signal is corrected and projected into the time-frequency domain through the TFR, achieving a clear separation of the modes of the received signal in the time-frequency domain. Based on different marine environments, a bandpass filter is designed to extract the modes. A depth estimation function is constructed to match the actual energy with the simulated signal energy, thus enabling target depth estimation based on the Warping transform under non-ideal waveguide conditions. Simulation experiments and real data experiments verify the method's

feasibility under various sound speed profiles. The effect of the signal-to-noise ratio on the method's accuracy is also discussed. Compared to the traditional Warping transform and the frequency-dispersive transform, the proposed method achieves a higher degree of mode separation, and the depth estimation results are more accurate under non-ideal waveguide conditions with variable sound speed profiles.

## Basic theory of warping transforms

Based on the normal modes theory, the acoustic field in an ideal waveguide can be expressed as a superposition of the normal modes of each order, and the steady phase method yields the acoustic field representation as follows:

$$p(r, t) = \sum_{m=1}^{N} a_m(t) e^{j2\pi f_{cm} \sqrt{t^2 - \left(\frac{r}{c_0^2}\right)}} \tag{1}$$

where $p$ is the acoustic pressure, $m$ is the mode number of the normal mode, $f_{cm}$ is the cutoff frequency of the $m$-th mode in an ideal waveguide, $a_m(t)$ is the instantaneous amplitude of the $m$-th normal mode, $r$ is the horizontal distance from the source to the receiver, and $c_0$ is the sound speed in the waveguide.

The time-domain Warping transform operator based on an ideal waveguide can be expressed as follows:

$$h(t) = \sqrt{t^2 + t_r^2} \tag{2}$$

In this case, $t_r = r/c_0$. Warping is a unitary transformation and the result of applying the Warping transformation $W_h$ to a function $y$ of variable $t$ is as follows:

$$W_h y(t) = \sqrt{|h'(t)|} y[h(t)] \tag{3}$$

This (Eq 3) represents two different forms of the same signal before and after the transformation, both containing the same signal energy, which supports the following depth estimation. The expression for the acoustic field after the Warping transformation is obtained by substituting Eq (2) into Eq (1):

$$p_h(t) = W_h p(t) = \sqrt{|h'(t)|} p[h(t)] = \sqrt{|h'(t)|} \sum_{m=1}^{N} a_m[h(t)] e^{j2\pi f_{cm} t} \tag{4}$$

The term $\sqrt{|h'(t)|}$ ensures that the signal energy remains unchanged before and after the transformation, and the transformed modes contain only the single frequency $f_{cm}$.

## Depth estimation method based on warping transform

The Warping transform utilizes the Warping operator to compensate for the phase. The traditional Warping transform is derived based on the Pekeris waveguide, which reduces the frequency aggregation effect of the Warping transform to a certain extent due to environmental influences and other factors in the actual underwater environment, thus affecting the accuracy of target parameter estimation. Therefore, this study corrects the Warping transform based on the BDRM model and proposes a depth estimation method for the target sound source under non-ideal waveguide conditions. Compared to the traditional Warping transform, the proposed method can be applied to the actual marine environment. The flow of the modified Warping transform design and the depth estimation method is shown in Fig 1.

The BDRM theory was proposed by Zhang and Li [29]. The group velocity and attenuation of the normal mode are defined by combining the cyclic distance of the intrinsic sound line

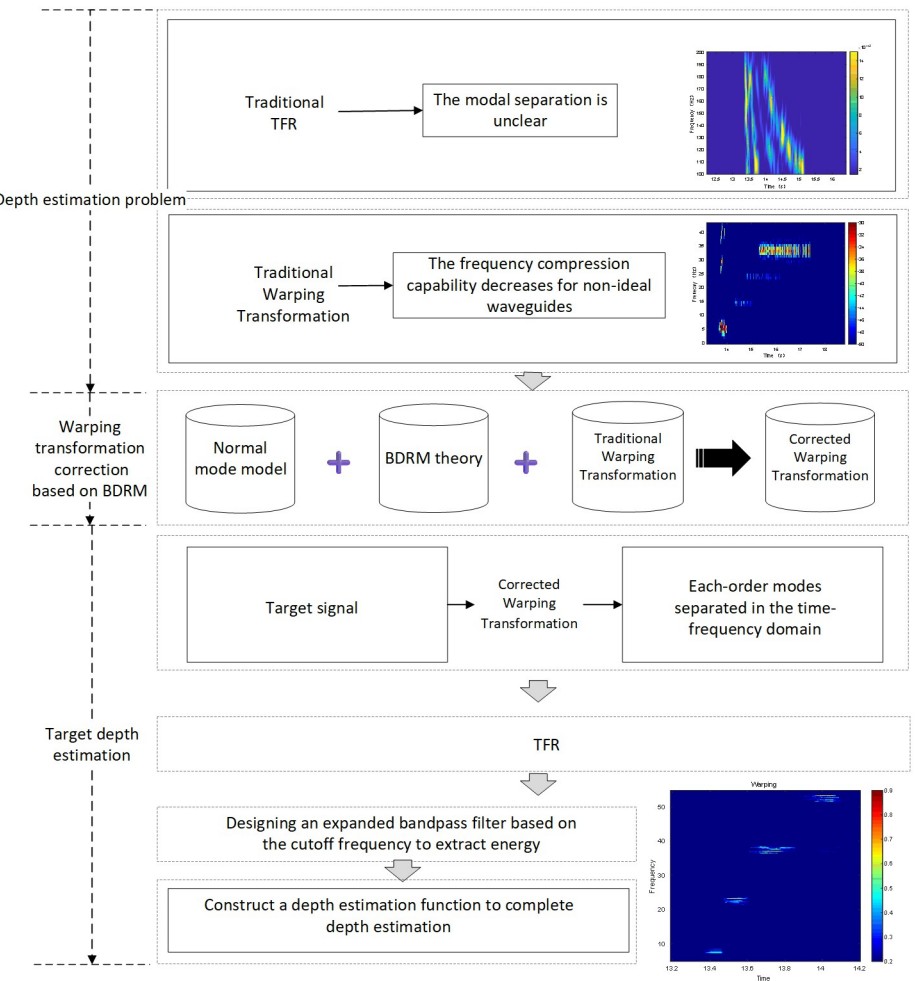

**Fig 1. Flowchart of depth estimation based on Warping transform.**

and the reflection coefficients. Based on the BDRM theory, with the correction of the Warping transform generalized to the non-perfect waveguide, the underwater sound velocity profile can be defined as follows:

$$c(z) = \bar{c}(1 - a(z)) \tag{5}$$

where $a(z)$ is a function of depth and satisfies $\int_0^H a(z)dz = 0$, so the relationship between the number of waves in the waveguide and the average wave number can be derived:

$$k(z) = \frac{\omega}{c(z)} \approx \bar{k}(1 + a(z)) \tag{6}$$

The BDRM model cyclic distance and eigenvocal line can be approximated as follows:

$$S(\mu_m) \approx \frac{2H\mu_m}{\sqrt{\bar{k}^2 - \mu_m^2}} \tag{7}$$

$$T(\mu_m) \approx \frac{2H\bar{k}}{\bar{c}\sqrt{\bar{k}^2 - \mu_m^2}} \tag{8}$$

where $H$ is the water depth and $\mu_m$ is the number of horizontal waves in the first $m$ horizontal wave number of the order waveguide, which can be expressed as follows:

$$\mu_m = \frac{\bar{k}r}{\bar{c}t(1+\varepsilon)} \tag{9}$$

Substituting the eigenequation at $t$ Taylor expansion and integration at the point gives the first $m$. The instantaneous phase of the order normal modes is:

$$\psi_m(t) \approx 2\pi\left[\frac{(2m-1)\bar{c}}{4H}\sqrt{t^2 - (r/\bar{c})^2} + \frac{\bar{c}\phi_b}{2H\pi}\sqrt{t^2 - (r/\bar{c})^2}\right] \tag{10}$$

For the SRBR normal modes, the seafloor reflection phase shift is $\varphi_b(t) = -2\phi_b$. Considering the seafloor absorption, the $\varphi_b(t)$ can be defined as follows:

$$\varphi_b(t) = -\arctan\left(\frac{2\rho_1\bar{c}\rho_2 c_2\sqrt{1 - \frac{r^2}{\bar{c}^2 t^2}}\left(\sqrt{\left[\frac{c_2^2}{\bar{c}^2\bar{c}^2 t^2} - (1-b^2)\right]^2 + 4b^2}\right)^{\frac{1}{2}}\cos\frac{\phi_s}{2}}{\rho_2^2 c_2^2\left(1 - \frac{r^2}{\bar{c}^2 t^2}\right) - \rho_1^2\bar{c}^2\sqrt{\left[\frac{c_2^2 r^2}{\bar{c}^2\bar{c}^2} - (1-b^2)\right]^2 + 4b^2}}\right) \tag{11}$$

$$\cos\frac{\phi_s}{2} = \sqrt{\frac{1}{2} + \frac{1}{2}\frac{\frac{c_2^2}{\bar{c}^2\bar{c}^2 t^2} - (1-b^2)}{\sqrt{\left[\frac{c_2^2}{\bar{c}^2\bar{c}^2} - (1-b^2)\right]^2 + 4b^2}}} \tag{12}$$

where $b$ is the seafloor absorption coefficient, $\rho_1$ is the density of the water column, and $\rho_2$ is the seafloor density, and $c_2$ is the seafloor sound velocity. Substituting Eq (10) into (1) yields the time-domain acoustic field expression based on the BDRM model as follows:

$$p(t,r) = \sum_{m=1}^{N} a_m(t)e^{j2\pi\psi_m(t)}$$
$$\approx \sum_{m=1}^{N} a_m(t)e^{j2\pi\left[\frac{(2m-1)\bar{c}}{4H}\sqrt{t^2 - (r/\bar{c})^2} + \frac{c\phi_b}{2H\pi}\sqrt{t^2 - (r/\bar{c})^2}\right]} \tag{13}$$

From Eqs (10) and (11), the representation of the sound field consists of two parts, and Eq (10) can be split into two parts:

$$\psi_m(t) = 2\pi[f_{cm}\xi(t) + \chi(t)] \tag{14}$$

Among them.

$$f_{cm} = \frac{(2m-1)\bar{c}}{4H} \tag{15}$$

$$\xi(t) = \sqrt{t^2 - \left(\frac{r}{\bar{c}}\right)^2} \tag{16}$$

$$\chi(t) = \frac{\bar{c}\phi_b(t)\xi(t)}{2H\pi} \tag{17}$$

Therefore, the Warping transform correction of the BDRM model must compensate for the phases of the two components, and the design of the Warping operator should also incorporate the $\xi(t)$ and $\chi(t)$ compensation of both parts should be considered. $\chi(t)$ Independent of the modal order, the compensation operator for $\chi(t)$ compensation operator can be designed as follows:

$$(\mathbf{W}_1 x)(t) = x(t)e^{-j2\pi\chi(t)} \tag{18}$$

$\xi(t)$ with the cutoff frequency $f_{cm}$ is closely related, for which the compensation and compensation operator can be designed as follows:

$$(\mathbf{W}_2 x)(t) = |w'(t)|^{\frac{1}{2}}x[w(t)] \tag{19}$$

$$h_{bdrm}(t) = \sqrt{t^2 + \left(\frac{r}{\bar{c}}\right)^2} \tag{20}$$

The final corrected Warping transform is obtained by combining the two correction operators into one:

$$(\mathbf{W}_{bdrm})(t) = \left|\frac{1}{h_{bdrm}(t)}\right|^{\frac{1}{2}} x[h_{bdrm}(t)]e^{-i2\pi\chi[h_{bdrm}(t)]} \tag{21}$$

$$p_{bdrm}(t) = \left|\frac{1}{h_{bdrm}(t)}\right|^{\frac{1}{2}} a_m[h_{bdrm}(t)]e^{\frac{j\pi(2m-1)\bar{c}-2\bar{c}\phi_b\sqrt{r^2+(r/\bar{c})^2}\sqrt{r^2+t^2}}{2H}} \tag{22}$$

The modified Warping transform of the received signal and the TFR can obtain the separated modes in the time-frequency domain. Similar to the traditional Warping transform, the modified Warping transform based on the BDRM model compresses the modes of the pulsed signals to a single-frequency signal near the cutoff frequency. Therefore, the actual waveguide parameters are combined to construct a simulated bandpass filter for the normal modes of the BDRM model to extract the energy of each order mode. The actual waveguide parameters are then utilized to construct an analog bandpass filter at the cutoff frequency of each mode, allowing the energy of each mode to be extracted. In practice, even after the modified Warping transform of the received signal, the compressed modes are approximate single-frequency signals rather than absolute single-frequency signals, and the energy of the modes is not concentrated in a single frequency but distributed around the cutoff frequency. As a result, the bandpass filter must be expanded accordingly. The center frequency is the cutoff frequency of the first-order mode $m$. The center frequency is the cutoff frequency of the $m$ where the center frequency is set as the cutoff frequency of the first-order mode $f_{cm}$. The passband width $bw$ is 10% of the signal bandwidth $\Delta f$ of the bandpass filter as a group of filters. The bandpass filters

are designed as follows:

$$H(f) = \begin{cases} 1, f_{cm} - 0.1\Delta f < f < f_{cm} + 0.1\Delta f, \; m = 1, 2, 3 \dots N \\ 0, \text{else} \end{cases} \quad (23)$$

Finally, the contrast function is constructed to normalize the extracted modal energies of each order of the target signal and match them with the modal energies of each order of the signal computed by the propagation model to achieve depth estimation of the target. The depth estimation contrast function is defined as follows:

$$G = 10 \log_{10} \left( \frac{M}{\Sigma \left( E_{m_r} - E_{m_s} \right)} \right) \quad (24)$$

where $E_{m_r}$ is the actual extracted energy of each order from the target and $E_{m_s}$ is the calculated energy of each order from the propagation model.

## Simulation experiment and analysis

The simulation experiments and analysis are divided into five subsections to verify the applicability of the proposed method under different sound speed profiles and the noise immunity of the method under different signal-to-noise ratios. Subsections 1, 2, and 3 are set up to verify the validity and superiority of the method under variable sound speed profiles. Subsection 4 is set up to verify the validity of the method in estimating the depth of the measured signal in the experimental sea area, and Subsection 5 is set up to investigate the effect of the signal-to-noise ratio on the method.

The experimental contents of each subsection are as follows: 1) Simulation experiment under the condition of shallow Pekeris waveguide, 2) Simulation experiment under the condition of waveguide based on a negative sound velocity gradient profile, 3) Simulation experiment under the condition of shallow waveguide in the Qingdao sea area, and 4) Numerical experiment of the sound source signal of an airgun in a sea area. The first three subsections compare the separation ability of the short-time Fourier transform, the frequency-dispersive transform, and the modified Warping transform for the modes of the impulse signals under different sound velocity profiles with a signal-to-noise ratio of 10 dB. The experiments compare the accuracy of the depth estimation of each method by repeating the tests. In addition, the depth estimation of the airgun sound source using the same three methods is conducted in the fourth subsection to compare the advantages and disadvantages of the three methods under measured marine data conditions. In subsection 5, under the simulated ocean environment conditions from subsection 3, the signal-to-noise ratio (SNR) is used as the only variable in repeated experiments to investigate the effect of SNR on the depth estimation results of the modified Warping transform. Except for the measured signal data in subsection 4, all the parameters of the marine environment and signal source are simulated using the Krakenc computational software.

### Pekeris waveguide simulation experiments

The simulation environment is modeled as shown in Fig 2. The water depth 100 $m$ and the density of the water body is 1.0 $kg/cm^3$. The underwater sound velocity is 1500 $m/s$. The density of the seafloor sedimentary layer is 1.8 $kg/cm^3$, and the sound velocity in the sedimentary layer is 1800 $m/s$. There is a vacuum above the top of the ocean. The center frequency of the signal is 150 $Hz$. The signal frequency ranges from 100–200 $Hz$, and the duration is 0.1$s$. The

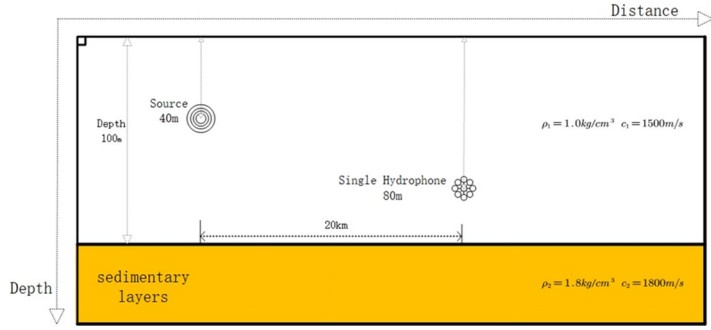

**Fig 2. Pekeris waveguide schematic of the simulated marine environment.**

depth of the sound source is 40 $m$. The hydrophone is placed 80 meters below the surface of the water, 20 $km$ away from the source. The depth search step is 0.1 $m$.

Based on the above simulation of the marine environment, the target signal is selected as a pulse signal in the $SNR = 10\ dB$ to accurately compare the advantages and disadvantages of the depth estimation accuracy of each method and reduce the influence of noise on the simulation. The target signal is simulated under the above conditions. Based on the theory of normal mode propagation, the higher-order modes have less influence on the acoustic field in the

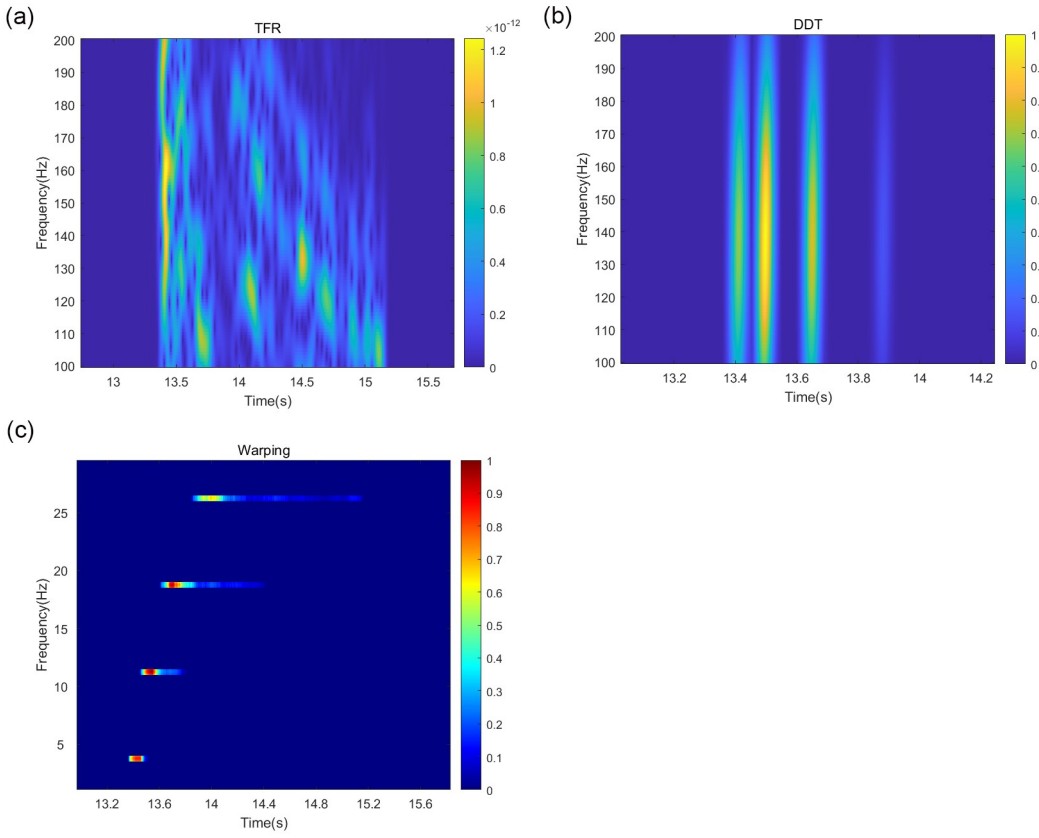

**Fig 3. Plot of modal separation results of different methods under Pekeris waveguide conditions.**

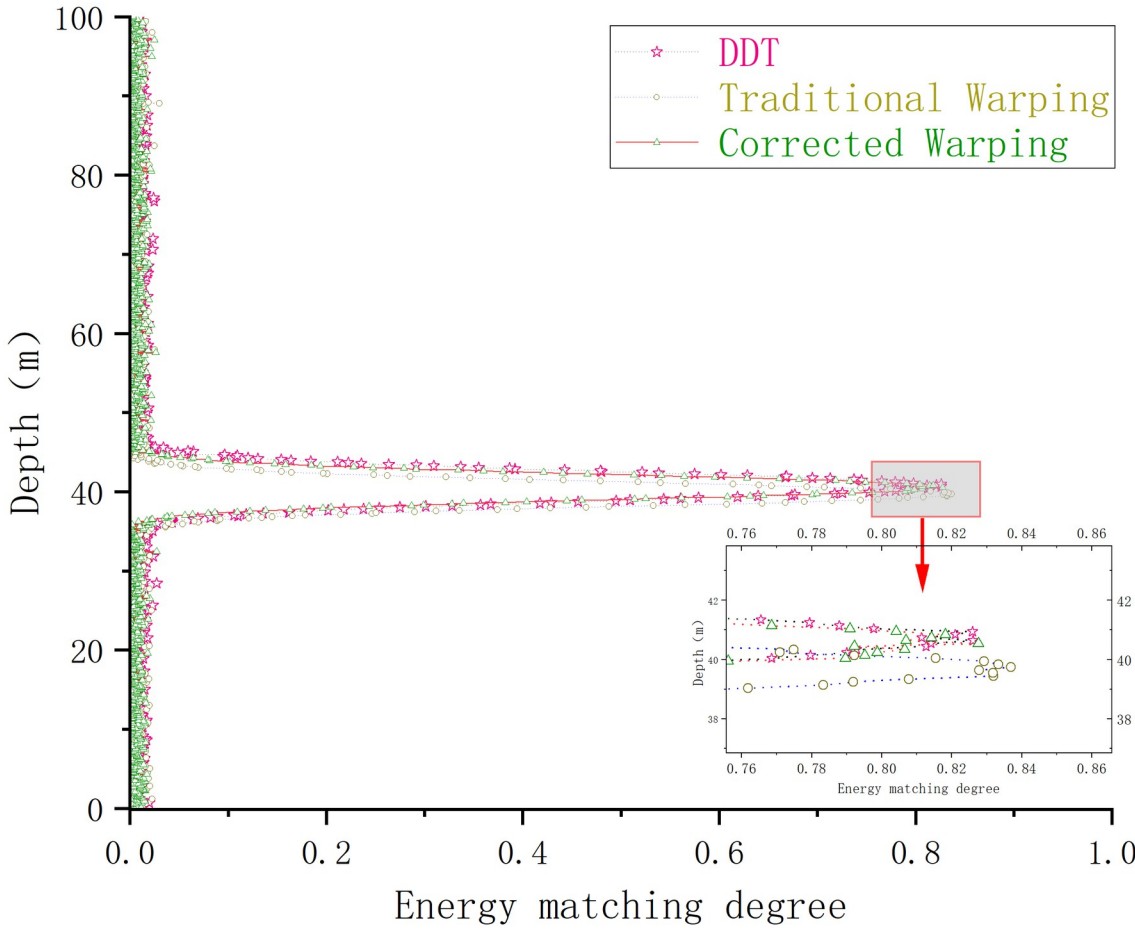

**Fig 4. Depth estimation maps of different depth estimation methods under Pekeris waveguide conditions.**

shallow sea environment, so the first four signal modes are selected for the simulation experiment. The TFR of the received signal after the short-time Fourier transform TFR, the frequency-dispersive transform TFR, and the modified Warping transform TFR filtered by the band-pass filter are shown in Fig 3.

A comparison of the sound source depth estimation results for each depth estimation method is depicted in Fig 4.

Under the Pekeris waveguide conditions, the conventional Warping, dissipative scattering transform, and modified Warping depth estimation results are 39.7, 40.7, and 40.6 *m*, with errors of 0.75%, 1.75%, and 1.5%, respectively. The errors are all within 5%, and all three methods provide a more accurate depth estimate of the sound source. The differences between the three methods are not significant under Pekeris waveguide conditions. The depth estimation was repeated 100 times for each of the three methods under constant environmental parameters to more accurately assess the validity and reliability of the three depth estimation methods and to avoid experimental errors. The success rate of the depth estimation results is defined as follows: since the waveguide environment is a Pekeris perfect acoustic field, a depth estimation result with an error greater than 3% is defined as an estimation failure, and the value is 0. A depth estimation result with an error of less than 3% is considered an estimation success, and

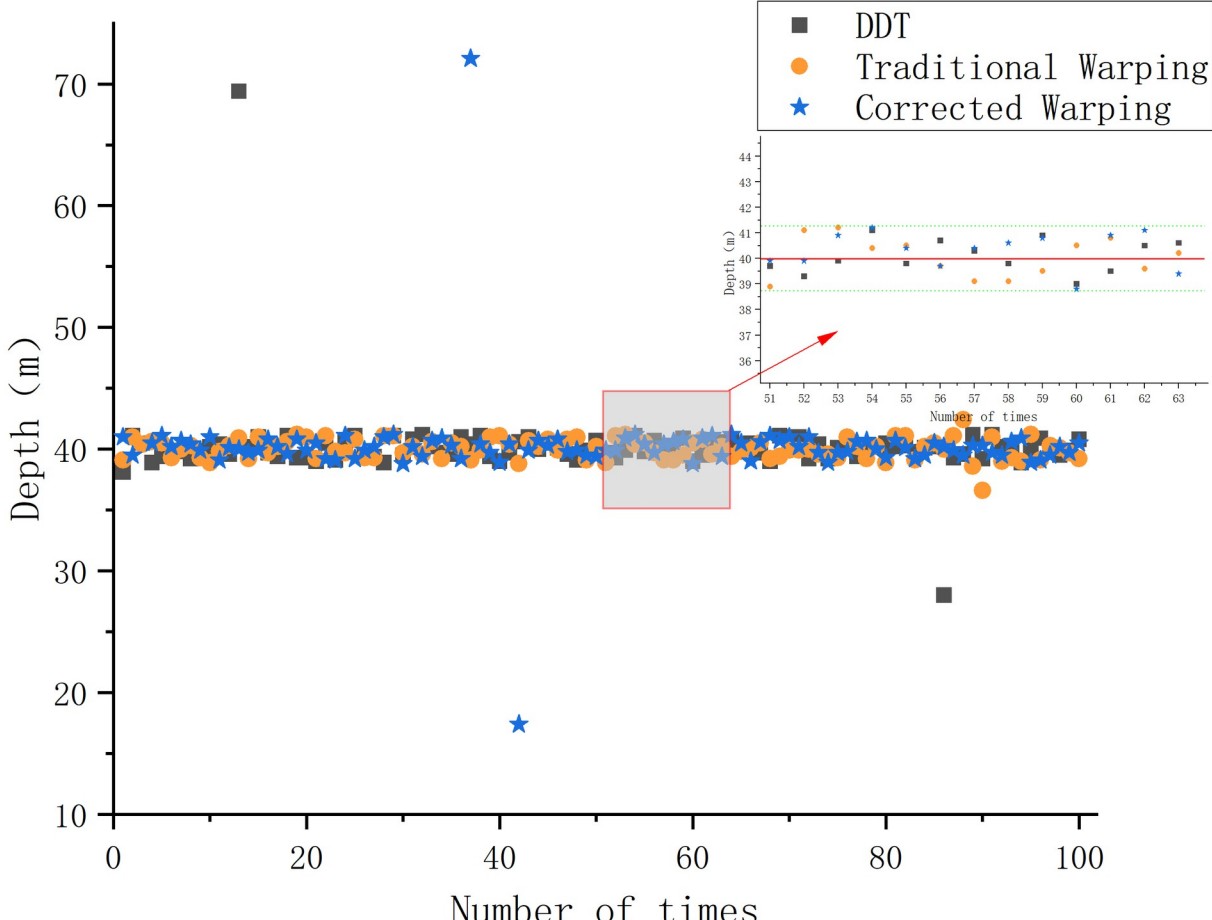

**Fig 5. Plot of the results of the repeated tests of different depth estimation methods under Pekeris waveguide conditions.**

the value is 1—the error rate. The average of 100 experiments is taken as the final success rate. The experimental results are shown in Fig 5.

Fig 5 visualizes the accuracy and the degree of error of the different methods in the repeated trials of depth estimation. The success rate of depth estimation under Pekeris waveguide

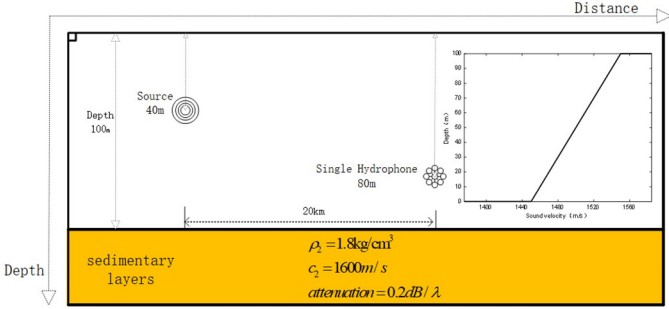

**Fig 6. Schematic of the simulated marine environment in the sea area with negative sound velocity gradient profile.**

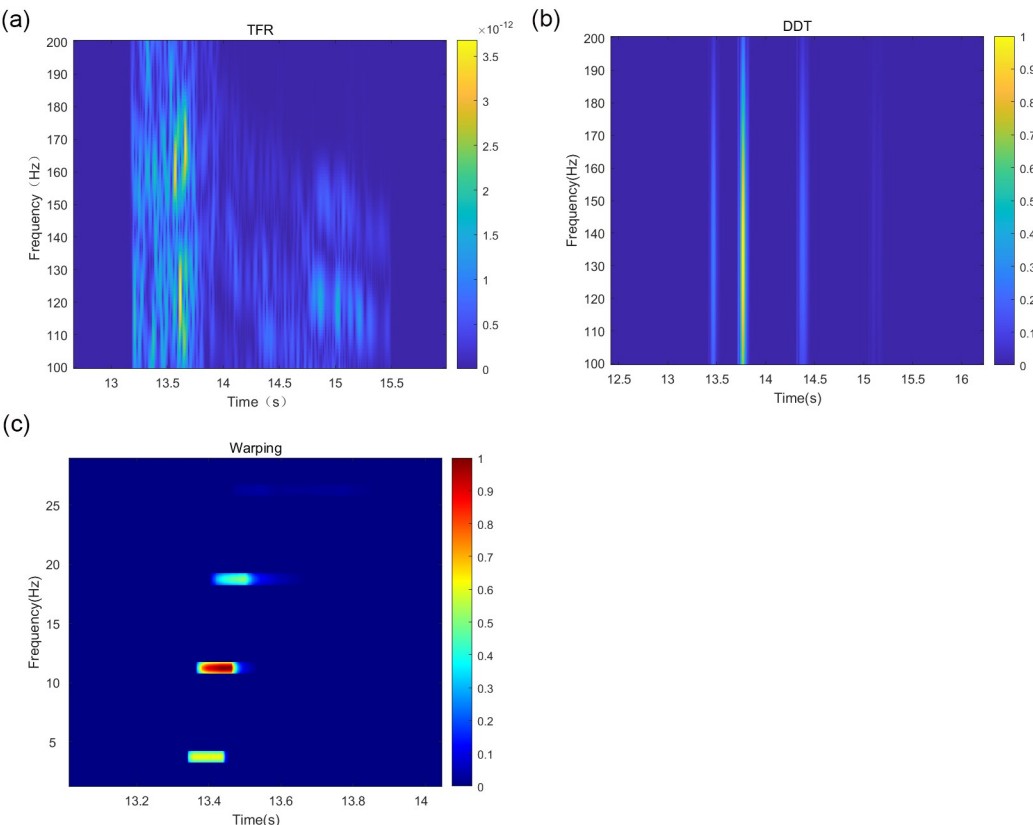

**Fig 7. Mode separation results of different methods under negative sound speed waveguide conditions.**

conditions is 97.8% for the conventional Warping transform, 96.1% for the DDT, and 96.4% for the proposed method.

Fig 5 and the experimental results indicated that, as verified by repeated experiments, all three methods demonstrate the ability to estimate the target depth under Pekeris waveguide conditions. The success rate of error depth estimation exceeds 95%, which indicates stability across all methods. The gap between the three methods under Pekeris waveguide conditions is small, and the DDT with corrected Warping does not reflect any significant superiority.

## Simulation of shallow-sea waveguide with negative gradient sound velocity profile

The simulation environment modeled in this subsection is shown in Fig 6, with the same water depth of 100 $m$. The density of the water column is 1.0 $kg/cm^3$, and the underwater sound velocity is provided in the sound velocity profile. The seafloor sedimentary layer density is 1.8 $kg/cm^3$, and the velocity of sound in the sedimentary layer is 1800 $m/s$. Above the top of the ocean is a vacuum. The signal has a center frequency of 150 $Hz$, a frequency range of 100–200 $Hz$, and a duration of 0.1 seconds. The sound source is positioned at a depth of 40 meters, and the hydrophone is placed at 80 m underwater, 20 km away from the sound source. The depth search step size is 0.1 m.

As in Section 3.1, other conditions remain unchanged in the above simulation environment for the simulation experiments. Under the condition of $SNR = 10$ $dB$, the first four modes of the received signal are taken for the simulation. The TFR of the received signal, after the short-

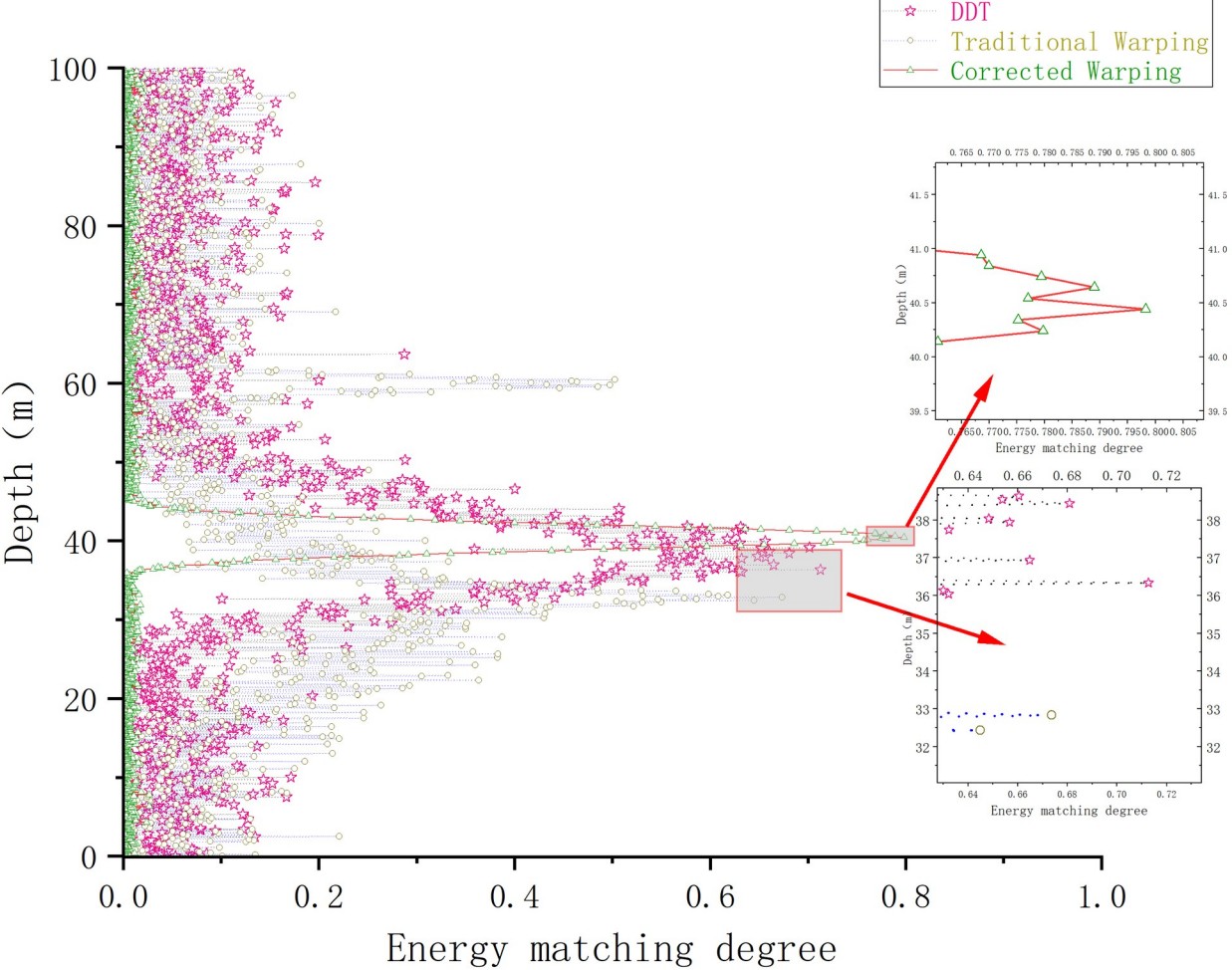

**Fig 8. Results of different Warping transform corrections for negative sound velocity gradient profiles.**

time Fourier transform TFR, the fading scattering transform TFR, and the modified warping transform TFR, are filtered by the band-pass filter and are shown in Fig 7, respectively:

Under the influence of the negative gradient sound velocity profile, the short-time Fourier transform is directly applied to the received signal, resulting in significant mode aliasing in the time-frequency domain. In contrast, the DDT and modified warping techniques achieve improved mode separation in the time-frequency domain under the same waveguide conditions. A comparison of the results from each depth estimation method for determining the depth of the sound source is presented in Fig 8:

Under the negative sound speed gradient shallow sea waveguide conditions, the conventional Warping, DDT, and modified Warping depth estimation methods yield estimates of 30.7, 36.4 and 40.4 $m$, respectively, with errors of 23.25%, 9%, and 1%, respectively. Compared to the Pekeris waveguide, the error rate of the depth estimation method based on the traditional Warping transform is significantly higher. In the non-Pekeris waveguide under the negative sound velocity gradient sound velocity profile condition, the traditional Warping transform produces larger errors and cannot achieve more accurate depth estimation results. However, the proposed depth estimation method maintains an error within 5%. Under these

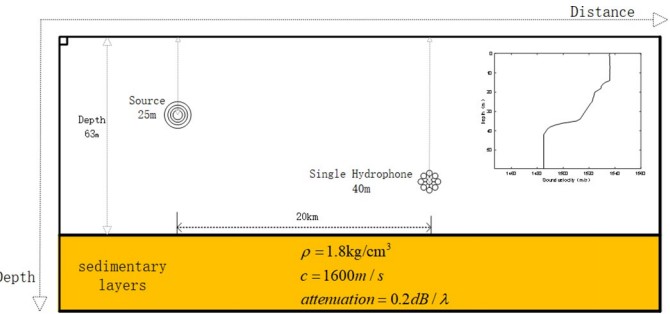

**Fig 9. Repeated test results of different depth estimation methods under negative sound velocity profile waveguide condition.**

experimental conditions, where the sound velocity varies uniformly with depth, the proposed method is not significantly affected by the negative sound velocity gradient sound velocity profile transformation. Therefore, the depth estimation results remain accurate. As in Section 3.1, the depth estimation is repeated 100 times using the three different methods. Because of

**Fig 10. Schematic diagram of simulated marine environment in Qingdao sea area.**

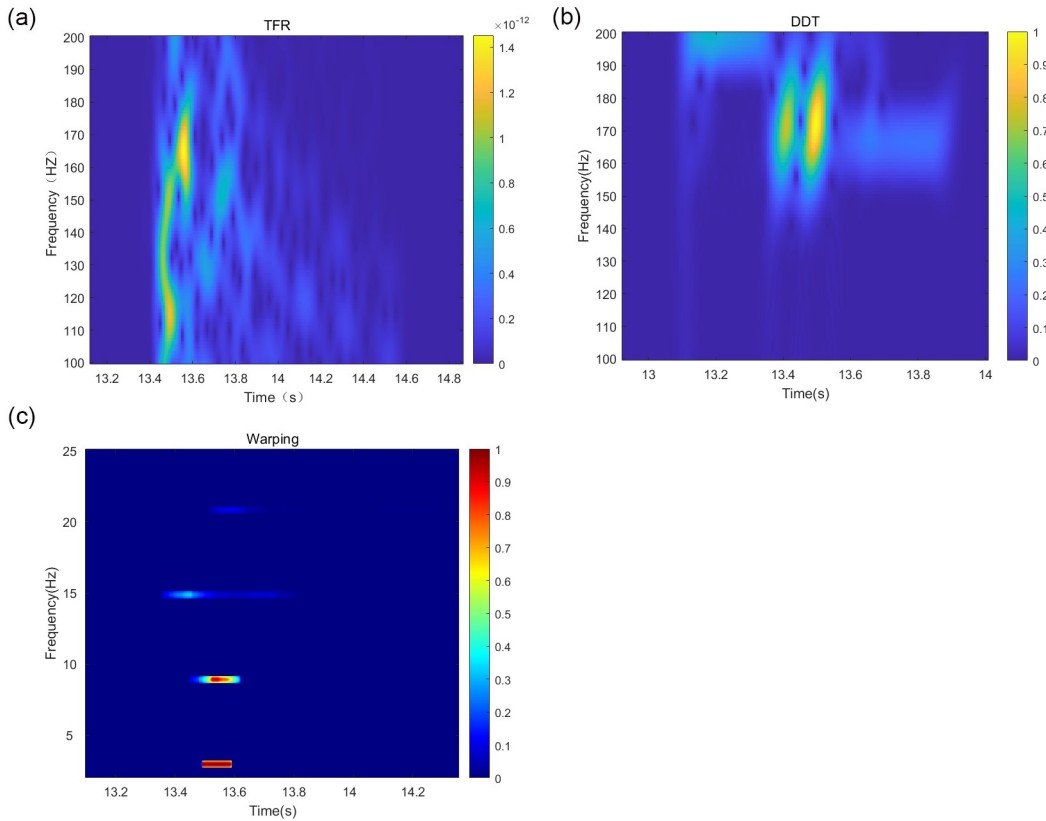

**Fig 11. Mode separation results of different methods in Qingdao shallow sea waveguide conditions.**

changes in the sound velocity profile, a depth estimation error greater than 5% is considered a failure. The results are shown in Fig 9.

Fig 9 exhibits that the conventional Warping and DDT depth estimation accuracies are significantly degraded compared to those observed in the Pekeris waveguide, and the errors increase. Under negative gradient sound velocity profile waveguide conditions, the success rate of depth estimation is 54.25% for the conventional Warping transform, 72.1% for DDT, and 97.2% for the method proposed in this study. The experimental results and Fig 9 show that the proposed method exhibits greater applicability and stability under the negative sound velocity gradient waveguide than DDT and the traditional Warping transform.

## Measured shallow sea waveguide in Qingdao sea simulation

The simulation environment model of the Qingdao Sea area is shown in Fig 10. The water depth is 63 $m$. The density of the water body is 1.0 $kg/cm^3$. The underwater sound velocity is shown in the sound velocity profile. The density of the sedimentary layer on the seabed is 1.8 $kg/cm^3$. The velocity of sound in the sedimentary layer is 1600 $m/s$. The seafloor attenuation is 0.2 $dB/\lambda$. The depth search step size is 0.1 m. There is a vacuum above the top of the ocean. The center frequency of the signal is 150 $Hz$. The center frequency of the signal *is* 150 $Hz$ within the range of 100–200 $Hz$. The depth of the sound source is 25 $m$. The hydrophone is placed 40 $m$ at a distance of 20 m from the source.

Similar to Section 3.2, the target signal is selected as a pulse signal, and the first four modes of the received signal are taken under the condition of *SNR* = 10 *dB*. The simulation considers

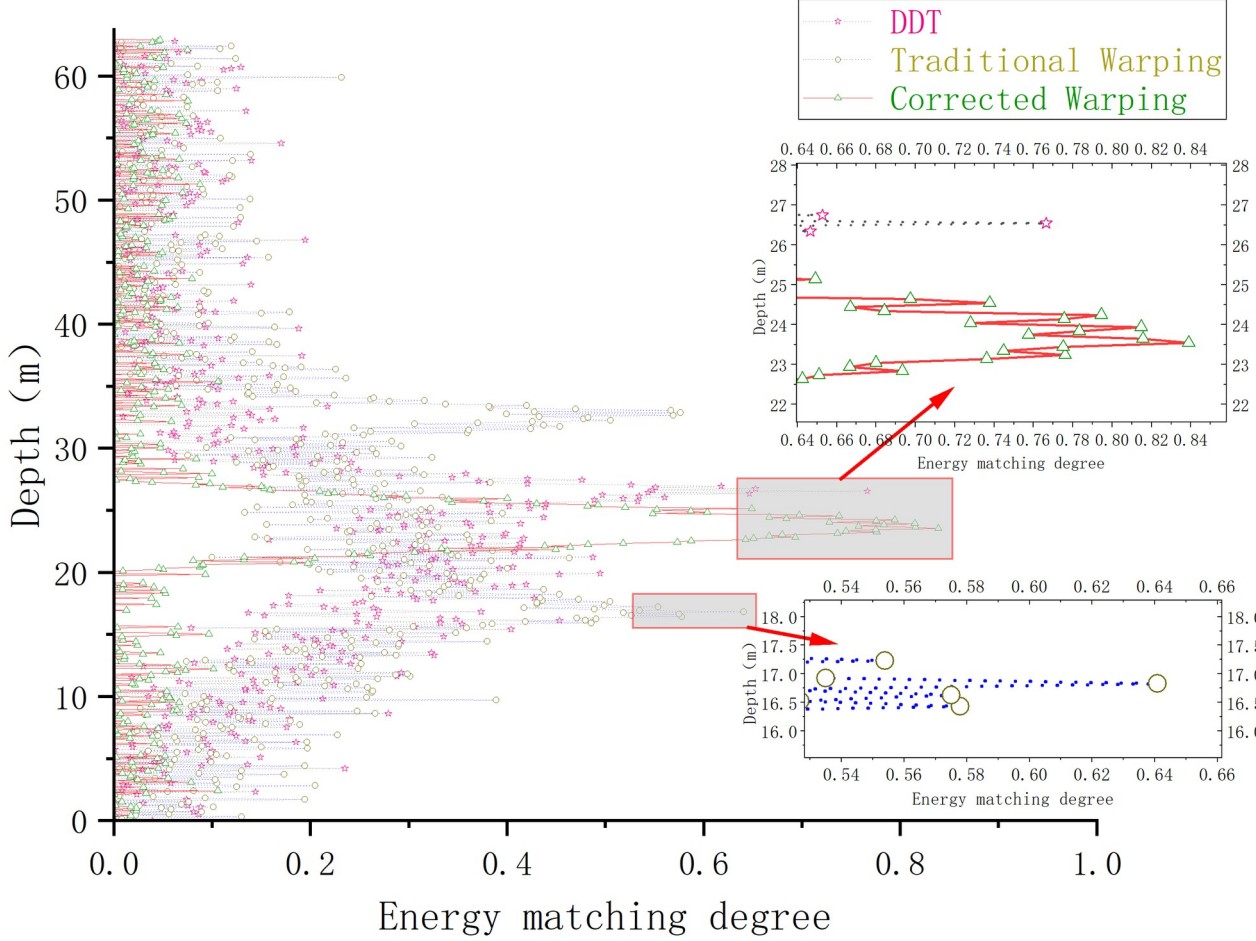

**Fig 12. Comparison of the depth estimation results of different methods under waveguide conditions in shallow waters of Qingdao.**

the first four orders of the modes of the received signal. The short-time Fourier transform, DDT, and modified Warping transform of the received signal, along with the TFR of the signal filtered by the band-pass filter, are shown in Fig 11, respectively:

Under the Qingdao shallow-sea waveguide conditions, the mode separation capabilities of the three methods degrade to different degrees, with the short-time Fourier transform and DDT being more significantly affected.

The depth estimation of the sound source, derived from the output of the depth estimation function, is illustrated in Fig 12.

Under the shallow waveguide conditions in the Qingdao Sea, the conventional Warping, DDT, and modified Warping depth estimation results are respectively 16.9, 26.5 *and* 23.8 *m*. The errors are 32.4%, 6%, and 4.8%, respectively. Compared to the Pekeris waveguide, the error rate of the traditional Warping transform remains larger in the non-Pekeris waveguide in the Qingdao shallow sea area, which cannot provide more accurate depth estimation results. Although the depth estimation results of the DDT are more accurate, its high-energy output range in Fig 12 is significantly widened compared to the experimental conditions in subsections 1 and 2. Although the corrected Warping transform is more accurate in the non-Pekeris waveguide conditions in the Qingdao shallow sea, the error of the depth estimation results

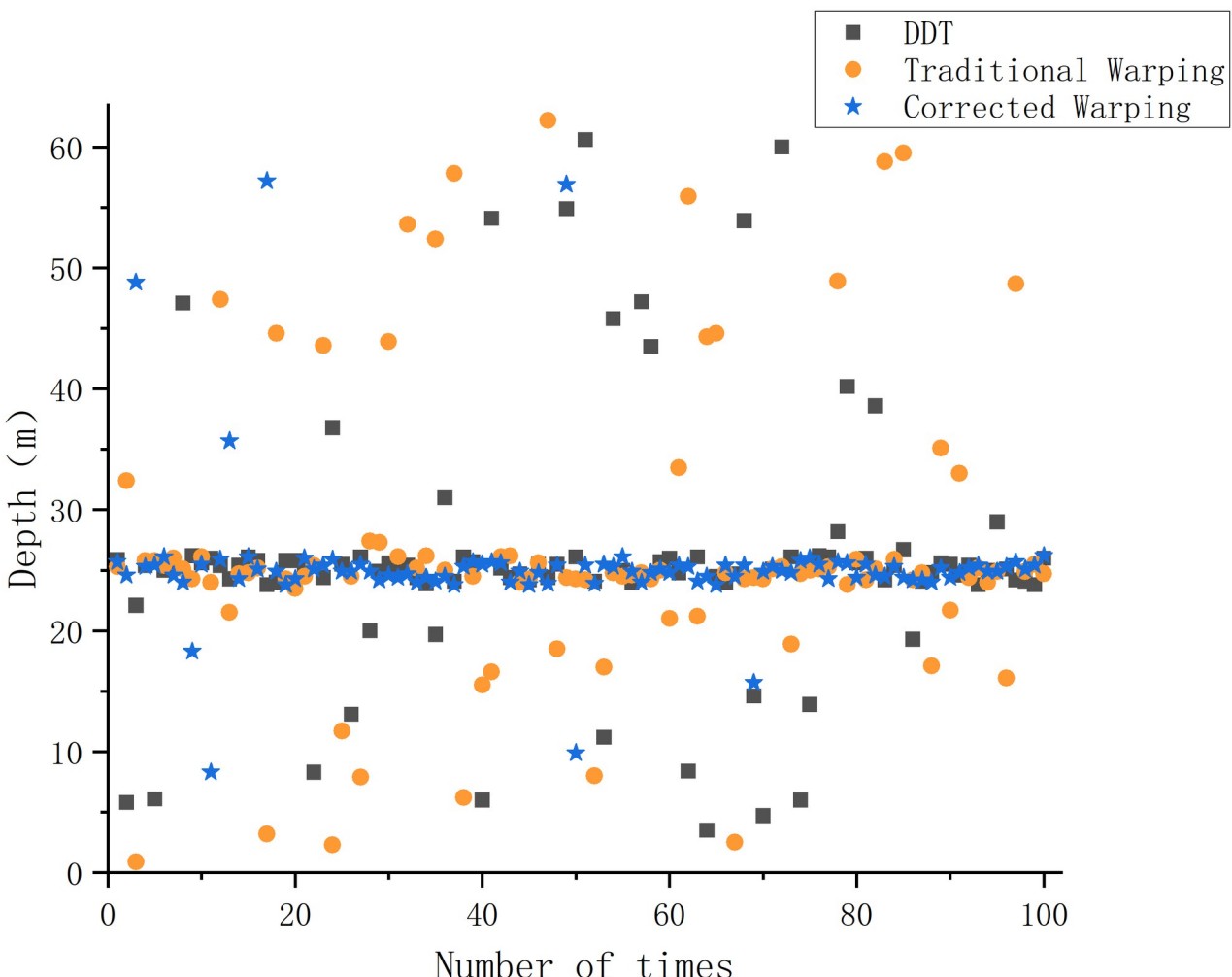

**Fig 13. Repeated test results of different depth estimation methods under waveguide conditions in Qingdao's shallow waters.**

remains within 5%. Similar to subsection 3.2, the three methods are repeated 100 times for depth estimation. Depth estimation errors greater than 5% are regarded as depth estimation failures, and the results are shown in Fig 13.

Under the Qingdao shallow-sea waveguide conditions, the success rate of depth estimation is 54.25% for the traditional warping transform, 68.8% for the DDT, and 95.1% for the method proposed in this study. Fig 13 and the experimental results indicate that the traditional warping transform remains sensitive to environmental parameter transformations under Qingdao shallow-sea waveguide conditions. The transform fails to separate the modes; the modal energy extraction remains incomplete, and the method cannot effectively estimate the target depth in the Qingdao Sea area. In addition, the DDT is more significantly affected under Qingdao real-time shallow-sea waveguide conditions, resulting in a widened error band. However, the modified warping transform based on the BDRM model achieves a depth estimation success rate of 95.1% under Qingdao shallow-sea conditions. This study highlights its superior applicability and stability under Qingdao shallow-sea waveguide conditions.

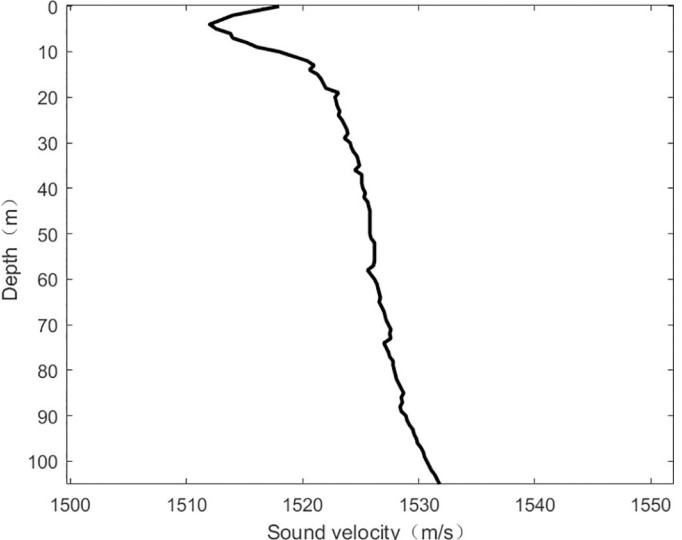

**Fig 14. Schematic of the sound velocity profile in the experimental sea area.**

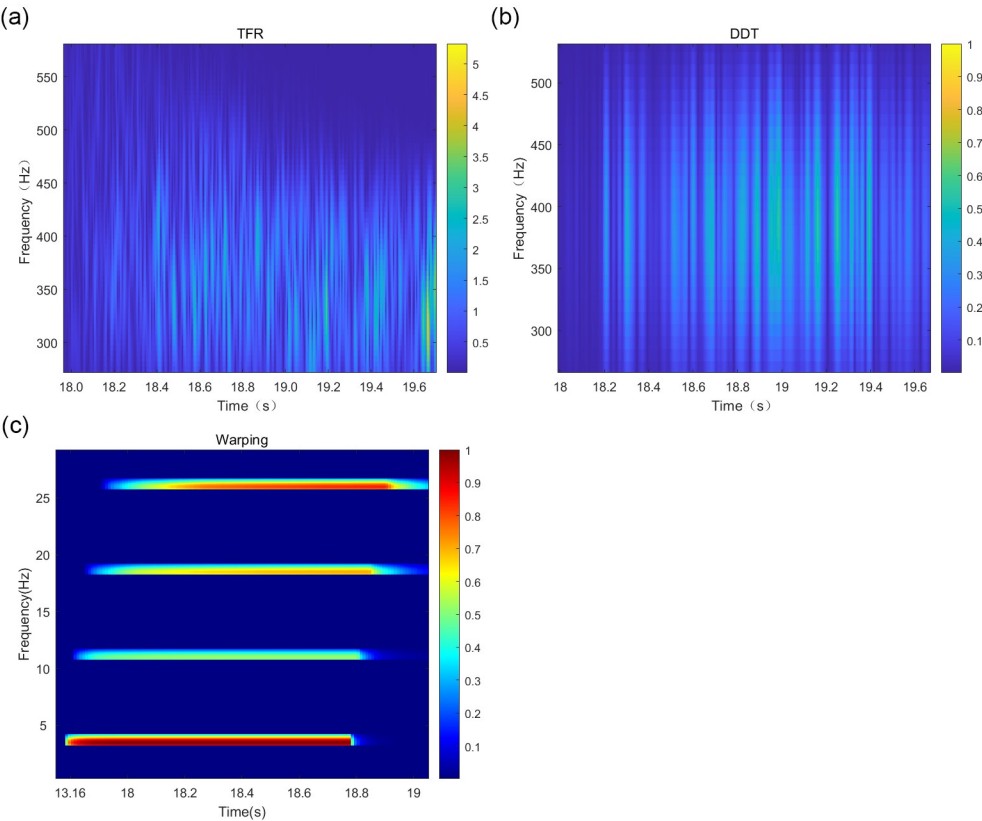

**Fig 15. Mode separation results of different methods in Qingdao shallow sea waveguide conditions.**

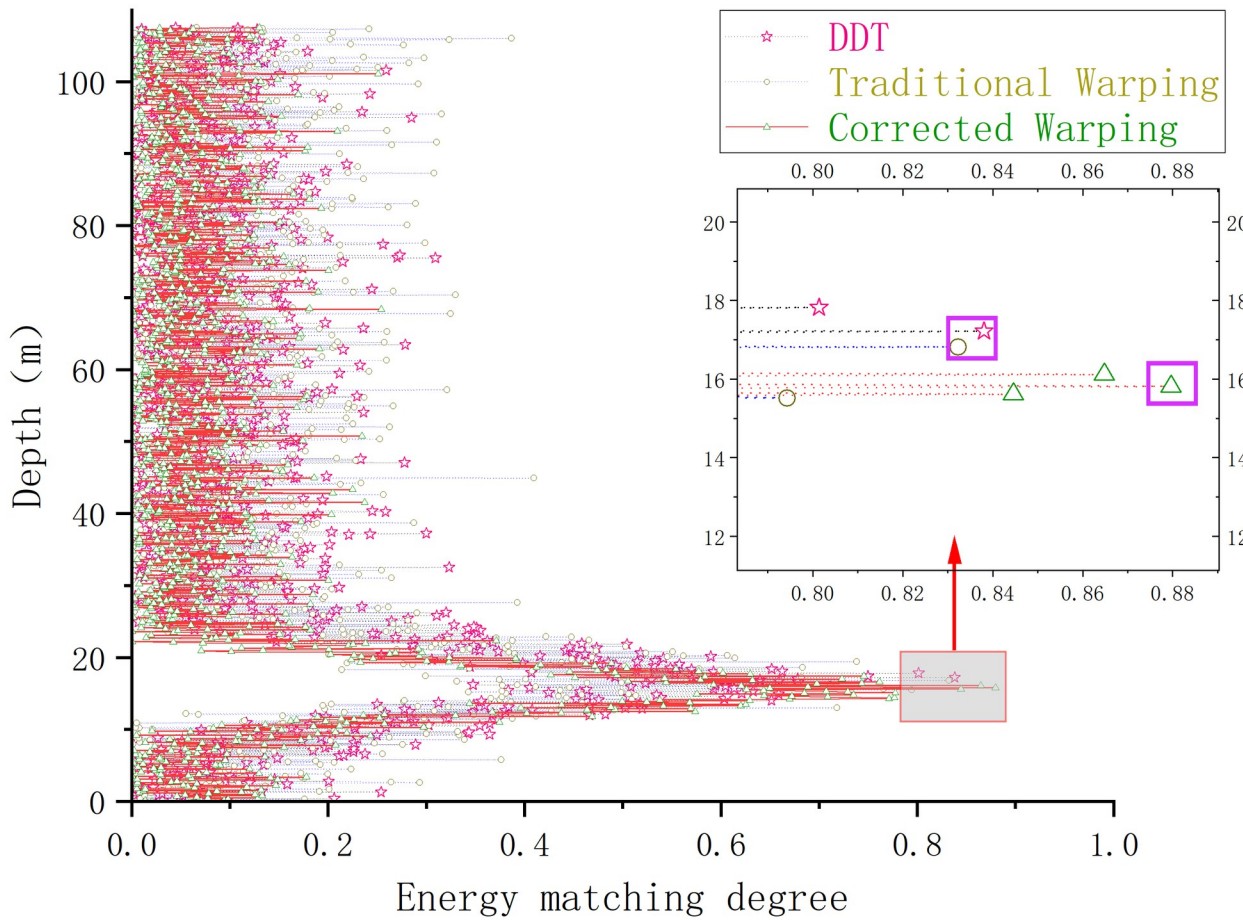

**Fig 16. Comparison of the results of different methods for estimating the depth of the sound source of measured airguns in the experimental sea area.**

### Numerical experiments of a sea trial of a measured explosive sound source signal

The experimental sea winter sound velocity profile (Fig 14) corresponds to a sea depth of about 107.5 *m* The explosion source is an airgun sound source at a depth of approximately 15 *m*. The receiving array consists of an 8-element vertical array, with the selected receiving array positioned at a depth of approximately 64.8 *m*. The distance to the receiving array is approximately 28.4 *km*.

The short-time Fourier transform, the DDT-corrected Warping transform, and the band-pass filtered signal TFR are shown in Fig 15.

The results of the depth estimation of the airgun sound source by each method are illustrated in Fig 16.

Fig 16 indicates that, in the numerical experiments using actual airgun data in the experimental sea area, the depth estimation results of the traditional Warping, the fading-dispersion transform, and the modified Warping methods are 17.2, 16.8, and 15.8 m, with errors of 14.6%, 12%, and 5.3%, respectively. Since the actual ocean sound velocity profile in the experimental area does not change significantly with depth and closely resembles the sound velocity profile of the Pekeris sound field, the estimation results obtained by the fading-dispersion

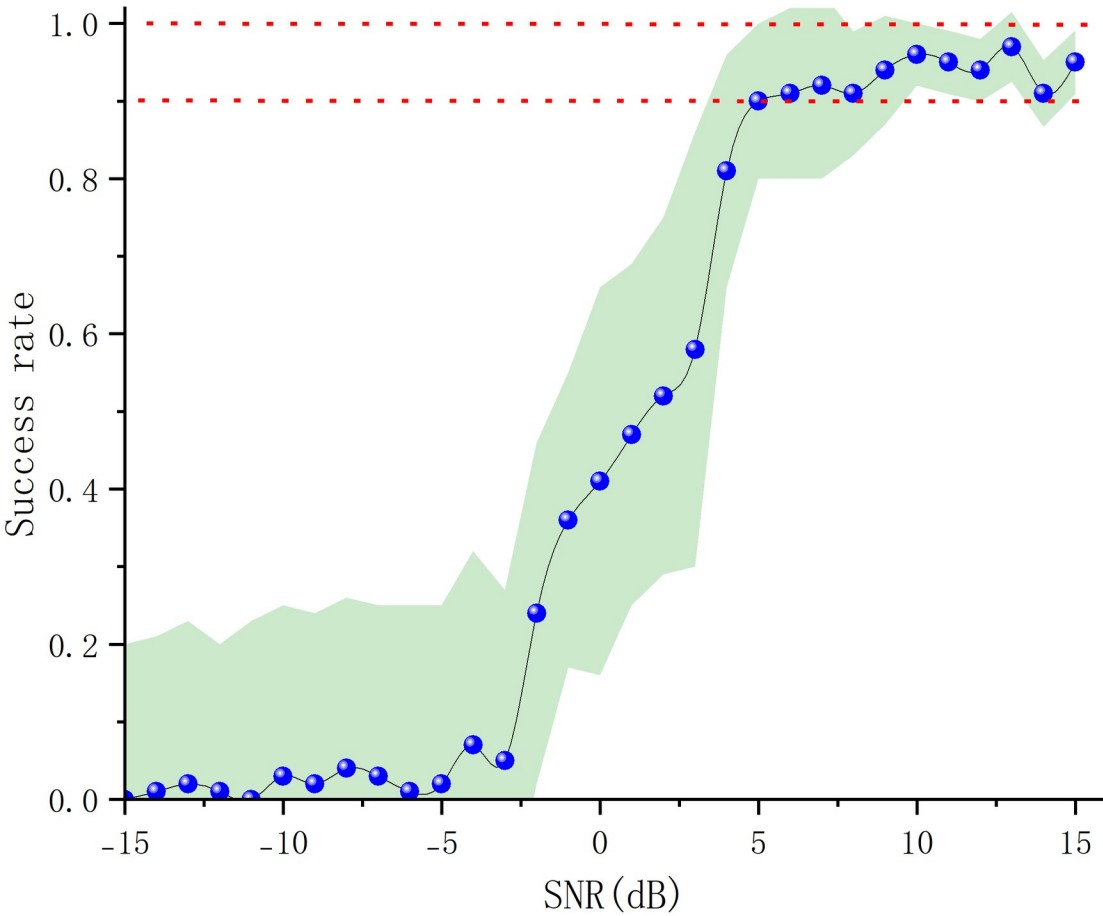

**Fig 17. Success rate of depth estimation for different signal-to-noise ratios.**

transform and the modified Warping transform remain acceptable. However, compared to these methods, the depth estimation achieved by the modified Warping transform proposed in this study is more accurate and proves more suitable for ocean environments where the sound velocity profile varies.

### An investigation of the effect of signal-to-noise ratio on methodology under measured shallow-sea waveguide conditions in Qingdao waters

The above experiments were conducted under a constant signal-to-noise ratio. The signal-to-noise ratio was set within the range of -15 dB to 15 dB under the marine environmental condition of 3.3 to more accurately explore the effect of the signal-to-noise ratio on the depth estimation accuracy of the modified Warping variational method. The signal-to-noise ratio was established as the experimental interval of the S/N ratio, with an interval increment of 1 *dB*. All other conditions were kept constant, and the depth estimation experiments were repeated 100 times for each S/N value at the specified interval points. Similarly, if the depth estimation result error exceeded 5%, the estimation was classified as a failure and assigned a value of 0. If the depth estimation result error was less than 5%, the estimation was considered successful and assigned a value of 1. The average of the 100 experiments was taken as the final success rate. The results of the experiments are depicted in Fig 17.

Fig 17 demonstrates that the blue dots represent the experimental success rates at different SNRs, while the green shaded area indicates the error range of localization accuracy during repeated experiments. It indicates that under conditions where the SNR is less than *-5 dB*, the depth estimation success rate of the method is extremely low, nearly 0, with significant errors. When the SNR exceeds *-3 dB*, the success rate of depth estimation begins to improve. Once the SNR surpasses *5 dB*, the success rate of depth estimation exceeds 90% and stabilizes, with the error range becoming smaller as the SNR increases.

## Conclusion

1. This study is based on the BDRM model and comprehensively considers the impact of sound speed profile variations on depth estimation. A depth estimation method using Warping transformation is proposed for single hydrophone applications. The method modifies the Warping transformation and reconstructs the sound field calculation formula to enable dual-phase compensation of the received signal, mitigating the effects of sound speed profile changes. The corrected Warping transformation is more suitable for non-Pekeris waveguides. The transformed received signal is then projected onto the time-frequency domain using TFR to separate its modal components. A bandpass filter is designed to match the waveguide environment, and a depth estimation matching function is constructed to estimate the depth of the target acoustic source. This approach provides a novel solution to the problem where Warping transformation is difficult to apply in non-Pekeris waveguides with varying sound speed profiles.

2. Simulation experiments verify the method's effectiveness and superiority. The method performs effective depth estimation of target pulse acoustic signals under various conditions: in the Pekeris waveguide, a simulated negative gradient sound speed profile waveguide, and the measured sea area waveguide in Qingdao with an SNR of 10 dB. It demonstrates a certain level of stability. The method achieves satisfactory depth estimation accuracy even at an SNR of 5 dB, and as the SNR increases, the error range decreases. In addition, experimental validation with actual airgun source data shows that the method can accurately estimate depth in real marine environments. This approach has broader applicability compared to traditional Warping transformations and DDT methods.

3. This study primarily investigates how to free the Warping transform from the constraints of the Pekeris waveguide and extends its application for depth estimation under a variable sound speed profile. The variation of the sound speed profile is the primary variable considered in this study; thus, the accuracy of sound speed profile data is crucial for the method's reliability. Mismatches in the sound speed profile directly impact the performance of the method. The study does not consider other marine environmental parameters, such as seabed phase shifts, sediment layer texture changes, ocean currents, bubbles, ice layers, and others, which were not included in the experiments. These factors must be incorporated in future research [35, 36], continuing to refine the Warping transform to better align with real-world marine environments. This study focuses on the proposal and improvement of depth estimation methods, selecting depth estimation accuracy and error values as the primary performance criteria. Other factors, such as the computational complexity of the method, are not considered here but will be addressed in the following studies. In addition, the proposed methods are based on SRBR simple waves, and further research is needed to explore how they can be generalized for broader applications.

## Supporting information

**S1 File.**
(RAR)

**S1 Data.**
(RAR)

## Author Contributions

**Data curation:** Du Siqi, Han Dong.

**Formal analysis:** Du Siqi.

**Investigation:** Li Sidi.

**Methodology:** Du Siqi.

**Resources:** Han Dong.

**Software:** Du Siqi, Liu Cong.

**Supervision:** Han Dong, Li Sidi.

**Writing – original draft:** Du Siqi.

**Writing – review & editing:** Du Siqi, Han Dong, Liu Cong, Li Sidi.

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
