## [Decision Letter · Decision Letter 0]

28 Nov 2024

PONE-D-24-47289

Underwater Target Depth Estimation:A Shallow-Water Broadband Acoustic Source Depth Estimation Method Based on Corrected Warping Transformation

PLOS ONE

Dear Dr. Han,

Thank you for submitting your manuscript to PLOS ONE. After careful consideration, we feel that it has merit but does not fully meet PLOS ONE’s publication criteria as it currently stands. Therefore, we invite you to submit a revised version of the manuscript that addresses the points raised during the review process.

We look forward to receiving your revised manuscript.

Kind regards,

Amit Kumar Goyal, PhD

Academic Editor

PLOS ONE

Journal Requirements:

3. We note that your Data Availability Statement is currently as follows: “All relevant data are within the manuscript and in Supporting Information files.”

Please confirm at this time whether or not your submission contains all raw data required to replicate the results of your study. Authors must share the “minimal data set” for their submission. PLOS defines the minimal data set to consist of the data required to replicate all study findings reported in the article, as well as related metadata and methods (https://journals.plos.org/plosone/s/data-availability#loc-minimal-data-set-definition). For example, authors should submit the following data: - The values behind the means, standard deviations and other measures reported; - The values used to build graphs; - The points extracted from images for analysis. Authors do not need to submit their entire data set if only a portion of the data was used in the reported study. If your submission does not contain these data, please either upload them as Supporting Information files or deposit them to a stable, public repository and provide us with the relevant URLs, DOIs, or accession numbers. For a list of recommended repositories, please see https://journals.plos.org/plosone/s/recommended-repositories. If there are ethical or legal restrictions on sharing a de-identified data set, please explain them in detail (e.g., data contain potentially sensitive information, data are owned by a third-party organization, etc.) and who has imposed them (e.g., an ethics committee). Please also provide contact information for a data access committee, ethics committee, or other institutional body to which data requests may be sent. If data are owned by a third party, please indicate how others may request data access.

5. Please remove your figures from within your manuscript file, leaving only the individual TIFF/EPS image files, uploaded separately. These will be automatically included in the reviewers’ PDF**.**

Reviewers' comments:

Reviewer's Responses to Questions

**Comments to the Author**

1. Is the manuscript technically sound, and do the data support the conclusions?

Reviewer #1: Partly

Reviewer #2: Partly

2. Has the statistical analysis been performed appropriately and rigorously? 

Reviewer #1: Yes

Reviewer #2: No

3. Have the authors made all data underlying the findings in their manuscript fully available?

Reviewer #1: Yes

Reviewer #2: No

4. Is the manuscript presented in an intelligible fashion and written in standard English?

Reviewer #1: No

Reviewer #2: No

5. Review Comments to the Author

Reviewer #1: The manuscript entitled "Underwater Target Depth Estimation: A Shallow-Water Broadband Acoustic Source Depth Estimation Based on Corrected Warping Transformation" has been written well but needs improvements. Authors need to improve these comments scientifically with proper references.

Suggestions for Improvement:

1. Include more quantitative comparisons with other state-of-the-art depth estimation techniques beyond traditional Warping transformation to strengthen the case for superiority.

2. Improve the resolution and labeling of figures (e.g., TFR plots and success rate tables) to enhance readability and interpretation.

3. Consider including experimental data from field trials to further substantiate the method's performance in practical scenarios.

4. Elaborate on the method's limitations, particularly regarding assumptions of known environmental parameters and potential challenges in obtaining these in real-world scenarios.

5. Expand the discussion on how the method could be generalized or adapted for broader applications, such as in elastic seabeds or under varying sound speed profiles.

6. Revise the manuscript for minor grammatical errors and improve the readability of complex sentences.

7. Ensure all cited works are correctly formatted and provide more details for entries like [25].

8. Include error bars or confidence intervals in tabular success rate data to illustrate variability in repeated trials.

Reviewer #2: The paper contains multiple basic errors. For instance, in equation (12), there is an incorrect use of parentheses, and the denominator is missing a square root term. Additionally, the density of seawater is incorrectly stated as 1.0kg\\/m^3 , whereas the actual value is approximately 1000kg\\/m^3 .

The concepts of propagation phase compensation and reflection phase compensation mentioned in the paper were already proposed in 2014, including similar ideas and simulation methods. Therefore, the paper lacks originality.

Niu H Q, Zhang R H, Li Z L. A modified warping operator based on BDRM theory in homogeneous shallow water[J]. Science China Physics, Mechanics and Astronomy, 2014, 57: 424-432.

There are several other issues in the paper. For example, while the use of filters is mentioned, the design of the filters is not described in the text. Additionally, the simulations were conducted only under a single condition of 10 dB, which lacks persuasiveness. Finally, the performance evaluation is limited to error and accuracy rates, with accuracy derived from the error metrics. The paper does not consider other evaluation criteria, such as computational complexity or other performance metrics.

6. PLOS authors have the option to publish the peer review history of their article (what does this mean?). If published, this will include your full peer review and any attached files.

Reviewer #1: **Yes: **Rajesh Mahadeva

Reviewer #2: No

---

## [Author Response · Author response to Decision Letter 0]

30 Dec 2024

Manuscript ID: PONE-D-24-47289

Article Type: Research Paper

Title: Underwater Target Depth Estimation:A Shallow-Water Broadband Acoustic Source Depth Estimation Method Based on Corrected Warping Transformation

Dear Editor 

Thank you very much for giving us a chance to revise our manuscript. The reviewer’s comments are valuable and very helpful for improving our research paper. Your professional perspective and insights not only demonstrate a deep understanding of the research field, but also inspire our subsequent work. We have carefully reformatted the manuscript and images in accordance with the journal’s requirements. The modifications made to the original version are highlighted in red font in the marked copy. The manuscript has been carefully proofread and edited by a professional language editor to improve clarity, grammar, and style. This editing process ensures that the manuscript adheres to the standards of academic writing and is free from linguistic errors. The editor is a native English speaker with expertise in scientific writing. We have carefully read all comments and have tried our best to revise the manuscript as per reviewers suggestions, which we hope to meet with acceptance requirements. 

Response to the reviewer’s comments

Reviewer #1

The manuscript entitled "Underwater Target Depth Estimation: A Shallow-Water Broadband Acoustic Source Depth Estimation Based on Corrected Warping Transformation" has been written well but needs improvements. Authors need to improve these comments scientifically with proper references.

1). Include more quantitative comparisons with other state-of-the-art depth estimation techniques beyond traditional Warping transformation to strengthen the case for superiority.

Response: We sincerely appreciate and have carefully considered your valuable suggestion. As a result, in the revised manuscript, we have included the Dispersive Dissipation Transform (DDT), an excellent time-frequency analysis method, in addition to the traditional Warping transform for comparison. In the introduction section of the manuscript, we have added a brief overview of the related research on DDT, including a description of its basic principles and capabilities (on page 5, lines 98 to 106), and we have cited the relevant references [21-24] in the manuscript’s reference section.

Furthermore, in Sections 1-4 of the simulation and analysis part, we have compared the ability to separate the modes of pulse broadband signals under various sound speed profiles and environmental parameters using traditional time-frequency methods (such as Short-Time Fourier Transform), DDT, and the modified Warping transform proposed in this manuscript. We have also added repeated depth estimation experiments using the DDT under the same conditions for the pulse source and compared the accuracy of depth estimation using all three methods under different waveguide conditions. Additionally, the original experimental result tables in the manuscript have been replaced with more intuitive line charts and scatter plots (e.g., Figures 4, 5, 8, 9, 12, 13, and 16) to demonstrate the superiority of the method proposed in this manuscript compared to other depth estimation methods, as well as its wide applicability in addressing variations in waveguide conditions.

2). Improve the resolution and labeling of figures (e.g., TFR plots and success rate tables) to enhance readability and interpretation.

Response: Respected reviewer thank you again for your valuable comments. Indeed, there was an issue with the resolution of some of the images in our manuscript. One of the reasons for this was a problem that occurred when we adjusted the manuscript format according to the journal’s requirements, which affected the image saving format. In this revised version, we have corrected the format for uploading images (e.g., the resolution of the TFR result in Figure 3).

Additionally, in response to your suggestion, in order to present our experimental results more clearly and intuitively, we have remade all the result analysis figures in the manuscript. We used line charts with clearer color and pattern contrasts (e.g., Figures 4, 8, 12, and 16). For important regions in the images, we added zoomed-in insets or prominent markers to improve their readability and interpretability. Furthermore, we replaced the original Tables 1 and 2 with scatter plots that clearly reflect the depth estimation accuracy and error of the various methods (e.g., Figures 5, 9, and 13), and similarly added local zoom-in views and distinct markers to key parts of the plots.

3). Consider including experimental data from field trials to further substantiate the method's performance in practical scenarios.

Response: We sincerely appreciate your valuable suggestion once again. This suggestion is very important to us, and adopting it will significantly enhance the persuasiveness of our manuscript. Due to the high costs of at-sea experiments and the complexity of experimental conditions (e.g., coordinating the deployment of research vessels), we were unable to organize a dedicated at-sea experiment for this manuscript within the limited time. However, we will aim to include such experiments in our future research.

As an alternative, we have included real experimental data from a winter sea trial conducted in a specific maritime area, where an airgun source was used. This data serves as field experimental data for numerical experiments to validate the feasibility of the proposed method. In Section 4 of the simulation and analysis part of the manuscript, we have provided a detailed description of the experimental conditions in the test area, and included a schematic of the measured sound speed profile (Figure 14). Using this measured data, we analyzed the depth estimation accuracy of the traditional Warping transform, DDT, and the modified Warping transform, and compared the modal separation abilities of the three methods on the real measured signals (Figure 15). The depth estimation results are clearly presented in Figure 16. This experiment, while validating the effectiveness of the proposed method, also highlights its superiority.

4). Elaborate on the method's limitations, particularly regarding assumptions of known environmental parameters and potential challenges in obtaining these in real-world scenarios.

Response: Respected reviewer thank you again for your comments. This is a very important suggestion, as it provides clearer requirements for the conditions under which the method can be applied. In the theoretical section and the derivation of the acoustic field equations of our manuscript, we have explained the environmental parameters required for the method. In each subsection of the simulation and analysis part, the assumptions regarding the environmental parameters are described in detail in the simulation environment section. Furthermore, in the conclusion section of the manuscript, in point 3 (from page 26, lines 550 to 565), we have clearly outlined the model conditions upon which the method relies, the importance of the accuracy of the sound speed profile, and its impact on the method, as well as the assumptions regarding other environmental parameters. Additionally, we have described the potential challenges in obtaining these environmental parameters and outlined the directions for future research improvements. Relevant references [35,36] have also been added.

5). Expand the discussion on how the method could be generalized or adapted for broader applications, such as in elastic seabeds or under varying sound speed profiles.

Response: Once again, we sincerely thank you for your valuable suggestions. The method proposed in this paper is well-suited for variable sound speed profile waveguides and can be applied to various depth-variable sound speed profile marine environments, provided that other environmental parameters are known and stable. Following your recommendation, we have also added a discussion in the conclusion section (from page 26, lines 555 to 559) regarding the future potential of the method. Specifically, we explored how the method could be extended to broader application scenarios. For example, incorporating the seabed phase shift parameter P could make the method applicable to elastic seabeds. We will also include this suggestion as part of our future research.

6). Revise the manuscript for minor grammatical errors and improve the readability of complex sentences.

Response: Dear reviewer thank you again for your valuable suggestions and comments. We sincerely apologize for the oversight in our manuscript, which led to some grammatical and spelling errors. In the newly submitted revised manuscript, we have carefully reviewed all the text and formulas to ensure their accuracy, and corrected spelling and missing errors in the images and formulas (e.g., in Equation (8) and Equation (12)).

To further improve the readability and fluency of the manuscript, we also submitted it to a professional native English speaker for a thorough check and correction of grammar and spelling. The editing report has been submitted along with the revised manuscript.

7). Ensure all cited works are correctly formatted and provide more details for entries like [25].

Response: We sincerely thank the reviewer for their careful reading and valuable corrections. We have corrected all the information related to reference [25], and in the revised manuscript, this reference is now cited as [29]. We have also rectified the formatting of all references to ensure they are correct and complete, and have made every effort to meet the publication requirements.

8). Include error bars or confidence intervals in tabular success rate data to illustrate variability in repeated trials.

Response: This suggestion is crucial to us. We realized that simply using result tables does not effectively and intuitively show the differences in depth estimation accuracy or the estimation errors. Therefore, we chose to replace the tables in the original manuscript with more intuitive scatter plots. Scatter plots not only visually display the variability in repeated experiments for a given method, but also provide a clear comparison with other methods, thus enhancing the readability of the results. Additionally, we have included error bands in the result figure (Figure 17) in Section 5 of the experimental and simulation part of the manuscript to highlight the variability in the repeated experiments.

Once again, we sincerely thank you for your hard work and valuable advice, which are crucial in enhancing the quality and depth of the paper. Your professional guidance not only helped us improve the paper, but also pointed the way for my future research. We look forward to the opportunity to seek your advice on more academic issues in the future and make joint progress.

Reviewer #2

1). The paper contains multiple basic errors. For instance, in equation (12), there is an incorrect use of parentheses, and the denominator is missing a square root term. Additionally, the density of seawater is incorrectly stated as 1.0kg\\/m^3 , whereas the actual value is approximately 1000kg\\/m^3 .

Response: We sincerely thank the reviewer for their careful reading and reminders. I apologize for our oversight. In the latest revised manuscript, we have corrected the errors in Equations (8) and (12), and have also fixed all incorrect seawater density units in the text and diagrams (e.g., Figures 2, 6, and 10).

Additionally, we have carefully reviewed the manuscript, corrected basic errors such as spelling mistakes, and, before submitting the final manuscript, we had it proofread and edited by a professional native English speaker to improve the grammar and sentence structure, ensuring that the research is conveyed as clearly and accurately as possible. We apologize once again for the errors in the original manuscript.

2). The concepts of propagation phase compensation and reflection phase compensation mentioned in the paper were already proposed in 2014, including similar ideas and simulation methods. Therefore, the paper lacks originality. Niu H Q, Zhang R H, Li Z L. A modified warping operator based on BDRM theory in homogeneous shallow water[J]. Science China Physics, Mechanics and Astronomy, 2014, 57: 424-432.

Response: We greatly appreciate the reviewer’s suggestions and the reference provided. We have carefully read this reference and incorporated it into the introduction and references section of our manuscript [31]. This paper has given us a deeper understanding of the combination of the BDRM model and the Warping transform from a fundamental theoretical perspective. By studying this paper and related content, we identified several shortcomings in our theoretical understanding of the BDRM research, which has been helpful for our subsequent studies. The referenced paper mainly compensates for nonlinear structure dispersion using a BDRM-based Warping operator, but we believe that there are some differences between the content and simulations in our manuscript and the work discussed in that paper.

The primary goal of our manuscript is to propose a more suitable depth estimation method for underwater targets, addressing the issue of reduced depth estimation accuracy in existing methods under varying sound speed profiles. In addition to performing biphasic correction on the Warping transform operator, we also reconstructed a new acoustic field expression based on the BDRM and Warping operators, as shown in Equation (22) of the manuscript, for use in the simulations. Furthermore, we introduced the TFR to achieve time-frequency domain separation of the various modal components of the normal wave. This approach utilizes the dispersion characteristics of the waveguide to extract effective information, rather than compensating for the dispersion of frequency-domain received signals. We also designed a simulated bandpass filter to extract the energy information of each mode and constructed an energy-matching depth estimation function to estimate the depth of the target.

Our research focuses on depth estimation for acoustic sources, aiming to develop a method that is more applicable to real-world marine environments and provides new ideas and options for future applications. This focus is reflected throughout the manuscript, including in the introduction (Chapter 1), method design (Chapter 2), experiments and simulations, and conclusion sections. In the simulations, we employed a richer set of sound speed profiles and real measured signals for repeatability and comparative experiments to validate the feasibility and superiority of the method. Additionally, the proposed method in the manuscript has continuity and potential for further improvement, laying a foundation for subsequent research and practical engineering applications.

3. There are several other issues in the paper. For example, while the use of filters is mentioned, the design of the filters is not described in the text. Additionally, the simulations were conducted only under a single condition of 10 dB, which lacks persuasiveness. Finally, the performance evaluation is limited to error and accuracy rates, with accuracy derived from the error metrics. The paper does not consider other evaluation criteria, such as computational complexity or other performance metrics.

Response: Dear reviewer thank you again for your valuable suggestions and comments.We sincerely thank the reviewer for raising these important issues, which are crucial and necessary for improving the readability and clarity of the manuscript. We apologize for the ambiguities and shortcomings present in the original manuscript. In response, we have addressed each of these issues in the newly submitted manuscript.

Firstly, in the section on the depth estimation method design (from page 12, lines 248 to page 13, lines 265), we have added a description of the filter design. This section now provides detailed information about the types of filters used, the design rationale, and the design process, along with the filter design formula (Equation 23).

Secondly, to address the concern regarding the lack of convincing results based on a single signal-to-noise ratio (SNR), we briefly explained in the experimental and simulation se

---

## [Decision Letter · Decision Letter 1]

12 Jan 2025

Underwater Target Depth Estimation:A Shallow-Water Broadband Acoustic Source Depth Estimation Method Based on Corrected Warping Transformation

PONE-D-24-47289R1

Dear Dr. Han,

We’re pleased to inform you that your manuscript has been judged scientifically suitable for publication and will be formally accepted for publication once it meets all outstanding technical requirements.

Kind regards,

Amit Kumar Goyal, PhD

Academic Editor

PLOS ONE

Additional Editor Comments (optional):

Reviewers' comments:

Reviewer's Responses to Questions

**Comments to the Author**

1. If the authors have adequately addressed your comments raised in a previous round of review and you feel that this manuscript is now acceptable for publication, you may indicate that here to bypass the “Comments to the Author” section, enter your conflict of interest statement in the “Confidential to Editor” section, and submit your "Accept" recommendation.

Reviewer #1: All comments have been addressed

2. Is the manuscript technically sound, and do the data support the conclusions?

Reviewer #1: Yes

3. Has the statistical analysis been performed appropriately and rigorously? 

Reviewer #1: Yes

4. Have the authors made all data underlying the findings in their manuscript fully available?

Reviewer #1: Yes

5. Is the manuscript presented in an intelligible fashion and written in standard English?

Reviewer #1: Yes

6. Review Comments to the Author

Reviewer #1: The authors implemented all comments suggested by me. Now, the manuscript is suitable for publication.

7. PLOS authors have the option to publish the peer review history of their article (what does this mean?). If published, this will include your full peer review and any attached files.

Reviewer #1: **Yes: **Rajesh Mahadeva

---

## [Editor Report · Acceptance letter]

15 Jan 2025

PONE-D-24-47289R1 

PLOS ONE

Dear Dr. Dong, 

I'm pleased to inform you that your manuscript has been deemed suitable for publication in PLOS ONE. Congratulations! Your manuscript is now being handed over to our production team.

Kind regards, 

on behalf of

Dr. Amit Kumar Goyal 

Academic Editor

PLOS ONE